# Performance of Co-Housed Neon Tetras (*Paracheirodon innesi*) and Glowlight Rasboras (*Trigonostigma hengeli*) Fed Commercial Flakes and Lyophilized Natural Food

**DOI:** 10.3390/ani11123520

**Published:** 2021-12-10

**Authors:** Robert Kasprzak, Anna Beata Grzeszkiewicz, Aleksandra Górecka

**Affiliations:** Department of Ichthyology and Biotechnology in Aquaculture, Warsaw University of Life Sciences (SGGW), Ciszewskiego 8, 02-786 Warsaw, Poland; anna.grzeszkiewicz1998@gmail.com (A.B.G.); olla125@gmail.com (A.G.)

**Keywords:** ornamental fish, aquaristics, histology, liver, digestive organs, skeletal double staining

## Abstract

**Simple Summary:**

The ornamental fish trade is a growing, developing global industry. However, on a scientific level, most aspects of aquarium fish nutrition remain to be a totally uncharted territory, awaiting to be explored. Thus, a feeding trial was conducted to compare the long-term effects of commercial flake feeds and dietary supplementation with natural food on the condition of neon tetras and glowlight rasboras, a setup which exemplified ordinary household community tanks. Even though there were no differences in growth between experimental groups, laboratory analyses revealed that the used feeding strategies had different outcomes in each species. Particularly, natural food appeared to cause a pathological lipid accumulation in the livers of rasboras, while no such effect was found in the tetras. The study highlights the need to conduct more in-depth feeding studies on ornamental fish, with special attention paid to their taxonomic origin and diversity.

**Abstract:**

Little to no research has been conducted thus far regarding aquarium fish nutrition. In order to ensure the welfare of house-kept ornamentals, such studies should take into account that there are distinct biological differences occurring between different fish species/taxa, especially in regard to the structure of their digestive organs. Accordingly, a 12-week trial was executed to assess the effects of two commercial flakes and a mix of lyophilized natural food on the condition of co-reared neon tetras, *Paracheirodon innesi* (Characidae), and glowlight rasboras, *Trigonostigma hengeli* (Danionidae). The four feeding groups were as follows: (T)—Tetra flakes; (O)—Omega flakes; (TO)—Tetra + Omega; (TOL)—Tetra + Omega + Lyophilizate (twice a week). There were no differences in final body weight (FBW) between the feeding groups of either species, but in the case of neon tetras, FBW increased significantly from the initial value only for the T group. However, histological observations and measurements of digestive organs (livers, intestines) showed pronounced differences between the two species. The supplementation with natural food in group TOL caused lipoid hepatic degeneration only in the rasboras. The healthiest histological structure of livers and longest intestinal folds were found in group T of the tetras and group TO of the rasboras. Whole-mount staining for bone and cartilage did not reveal any significant deformities or differences in terms of bone mineralization. In conclusion, it was outlined that concurrent feeding of co-housed, anatomically diverse ornamental fish species is a highly ambiguous task, because the nutritional strategy applied for a community tank may yield radically divergent effects, most of which may remain unnoticed when depending only on external body observations and measurements. Most emphatically, this was highlighted in regard to the dietary supplementation with natural food—although no significant effects were observed in neon tetras, severe lipoid liver degeneration occurred in glowlight rasboras.

## 1. Introduction

The ornamental fish trade is a vast market of near-global range which generates an estimated yearly income surpassing tens of billions euro, with freshwater species accounting for more than 90% of the trade [1,2]. These numbers obviously pale in comparison with the sizes of both the aquaculture production and capture fisheries [3], but it is a distinct and noteworthy industry nonetheless. It is, therefore, surprising that there is little to no scientific information available in regard to various aspects of aquarium fish keeping, such as dietary preferences or recommended feeding regimes for different species [4].

Feeding is arguably the most important variable in the course of fish rearing, both from a commercial and animal welfare standpoint. Therefore, the composition and contents of formulated inert diets are being dissected thoroughly in aquaculture studies, in a species-specific manner, with the main focus being placed on protein and lipid sources [5,6,7,8,9,10], but even advanced additives such as nucleotides [11], prebiotics [12] and probiotics [13] are in the scope of investigation. Furthermore, means to improve natural food, which is crucially important during early weaning, are being studied profoundly [14].

In comparison, the majority of scientifically verified reports about aquarium fish nutrition also concern the application of natural food for larval stages [15,16,17,18,19,20,21,22,23,24], but there are mostly vague reports about diets for fish which already underwent larval metamorphosis [25,26,27,28,29,30,31,32,33,34,35]. More attention has thus far been paid only to the Siamese fighting fish, *Betta splendens* [36,37,38,39,40,41,42,43,44], commonly known as the “betta”. For this species, natural food in the form of mosquito larvae (*Culex* sp.) apparently improves its coloration [39,43]. Furthermore, best fecundity was attained when using at least 50% of *Tubifex* sp. larvae in their diet, in expense of artificial feed [44], and when providing two meals per day (growth rate was improved, too) [36]. Similar results (better feeding efficiency and breeding indices) were shown when using *Spirulina* sp. as a supplementary ingredient in formulated pellets for poecilid livebearers [32,33]. In addition, a highly diversified mixture of natural and artificial food proved to be the most preferable for the green swordtail, *Xiphophorus hellerii* [27], while different live feed were shown to be an upgrade from commercial flakes when fed to the guppy, *Poecilia reticulata* [35]. Finally, nutritional studies on the model species, the zebrafish *Danio rerio* (Danionidae), have shown that intensive rearing with only *Artemia* sp. may drastically accelerate fish growth and maturation [45], and HUFA-enrichment of *Chironomidae* sp. larvae also improves fecundity [46]. However, well-balanced artificial diets can be highly beneficial for the overall performance of zebrafish, too [47,48,49,50] and dietary fish meal should not always be fully replaced with live food [51]. All in all, these results generally point toward a presumption that natural food is highly desirable in the nutrition of aquarium fish, especially when used in addition to formulated feed. Unfortunately, apart from some of the aforementioned studies on the betta [40,41,42,43] and zebrafish [45,47,48,51], most of the scientists cited above [26,27,28,29,30,31,32,33,34,35,36,37,38,39,46,49,50] did not use almost any analytical methodology other than standard growth parameters. Thus, they did not take into account that there are distinct internal biological features which distinguish the studied species, which may be of consequence during their co-housing in community tanks.

In terms of taxonomy, ornamental fish are a highly heterogenous collective. Nevertheless, they are classified together in ambiguous groups (such as herbi-, omni- or carnivores) by commercial feed producers [52], with disregard for essential differences in their anatomy, morphology and physiology. To outline an example, many popular, small freshwater species (<6 cm total body length) belong to two teleost families—the Characidae (“tetras”) and the Danionidae (“danios”, “rasboras”). Both families are classified, not-so-distantly, within the series Otophysi [53], but they differ in regard to the structure of their alimentary tracts, as danionids possess highly specialized pharyngeal teeth instead of typical maxillary teeth [54,55] and do not have a typical stomach (they are agastric) [56].

The neon tetra, *Paracheirodon innesi* (Myers, 1936) is a South American characid and is one of the most well-established and commercially important species in the aquarium hobby [57]. In fact, breeding protocols were already developed years ago, including actual scientific research [58,59,60,61]. Meanwhile, some relatively basic feeding-oriented studies were conducted on the species [29,30,31,62,63,64,65,66,67] and its congener, the cardinal tetra *Paracheirodon axelrodi* (Schultz, 1956) [34], but only standard body morphometrics were reported, without providing any insight into the structure and condition of internal organs.

The glowlight rasbora, *Trigonostigma hengeli* (Meinken, 1956) is an ornamental danionid species from Borneo and Sumatra [68]. It is by far not as popular as *P. innesi*, or even its own congener—the harlequin rasbora, *Trigonostigma heteromorpha* (Duncker, 1904)—both of which are already recognized as highly domesticated species [69]. Truly, apart from its original taxonomic description and reports about its natural populations, this species has not been the object of any published scientific experimentation.

When assessing the effects of feeding in aquaculture studies, the histology of the digestive organs is naturally one of the major features which is being looked at [70,71,72]; however, skeletal performance (mineralization) and feeding-induced deformities are also often described in various farmed fish [73,74]. While some of the aforementioned papers included relatively advanced analytical methods, such as digestive enzyme activity [41,42,43], histology has never been used before in feeding-related studies on ornamentals, with the exception of the zebrafish [45,51]. Skeletal development of aquarium fish is also a sparsely discussed topic. Thus far, it was only shown that low dietary phosphorus content results in vertebral column deformations in the guppy [75], while the deficiency of vitamin C causes comparable pathologies in juvenile oscars, *Astronotus ocellatus* (Cichlidae) [76]. Deformities also appear in zebrafish stocked at too high densities [77], but early transition from *Artemia* sp. to micro-diets [48] and adequate dietary phospholipid sources [47] reduce the occurrence of such anomalies.

Therefore, to improve the state of knowledge about this little-studied topic of aquarium fish nutrition, an experiment was designed to imitate a simple-yet-typical, two-species setup found in home aquaria, with *P. innesi* and *T. hengeli* used as representatives of the characid and danionid families, respectively. The main aims of the study were (1) to evaluate the effects of two commercial flake diets on the condition of co-housed, anatomically disparate ornamental fish and (2) to challenge the legitimacy of the common habit of aquarists to supplement formulated diets (flakes and pellets) with natural food. The ensuing laboratory analyses involved regular histological analyses of the digestive organs (including morphometrics of liver cells and intestinal folds), as well as the assessment of skeletal performance using whole-mount double staining for bone and cartilage.

## 2. Materials and Methods

### 2.1. Fish and Experimental Setup Preparation

Sub-adult specimens (~2 cm total length) of both neon tetra and glowlight rasbora were purchased from a local wholesaler and quarantined for two weeks in a 100 L tank, with highly diversified food provided twice daily, *ad libitum*. The pool of administered food included four different commercial flake products, as well as spirulina, frozen brine shrimp and frozen bloodworms. Afterwards, all fish were weighed (body mass, BM) and 16 specimens of each species were euthanized in 0.5‰ MS-222 solution (Sigma Aldrich Co., St. Louis., MO, USA) to provide a point of reference sample (named “Initial” group) for laboratory analyses.

The experimental setup was constituted by the co-stocking of remaining quarantined fish into eight 20 L aquaria, with eight tetras (mean initial body weight [IBW] = 272 ± 49 mg) and eight rasboras (mean IBW = 227 ± 67 mg) per each tank. In order to model the conditions of regular home aquaria and to minimize inter- and intraspecific antagonisms, four different artificial plants were placed in every tank and the bottom was lined with basalt substrate. Each aquarium was equipped with its own 25 W thermostat heater and cascade filter. The maintained temperature was 25 ± 0.5 °C and other water parameters were monitored daily using ProAquaTest Easy 7in1 test strips (JBL GmbH & Co. KG, Neuhofen, Germany). Tank water was replaced once daily (20% of the volume) with freshly demineralized water in order to keep its parameters within the following ranges: pH ≈ 6.4, nitrates <50 ppm and nitrites <10 ppm. A 10 h:14 h artificial photoperiod was provided throughout the entire study (ceiling light was switched on at 8:00–18:00).

### 2.2. Feeding Experiment

A duplicate of tanks with the co-stocked tetras and rasboras was assigned to one of four experimental feeding groups, with food being offered twice daily (at 9:00 and 16:00) and the daily rations equaling ~3–4% body weight (slightly exceeding the maintenance feeding requirement suggested for neon tetras [29] and zebrafish [78]). Non-eaten particles and feces were siphoned out after every feeding. Three different feeds were used in the study: TetraMin Flakes (Tetra GmbH, Melle, Germany), Omega One Freshwater Flakes (Omega Sea LLC, Painesville, OH, USA) and a pulverized lyophilizate mix of natural food (Katrinex, Sosnowiec, Poland). The basic proximate composition of the feeds (protein, fat, fiber, ash and moisture) was analyzed according to AOAC methods [79]. The ingredients and compositions of the feeds are shown in Table 1.

The four experimental feeding groups in the study were as follows:Group T was only given the Tetra flakes;Group O was only given the Omega flakes;Group TO was alternately given the Tetra and Omega flakes;Group TOL was alternately given the Tetra and Omega flakes, but also the lyophilizate mix (only twice a week, instead of one of the flake meals).

The detailed feeding schedule is outlined in Table 2. This feeding regime was chosen to represent a typical approach of hobbyists, who usually attempt to diversify the diets of their pets by alternating different commercial flakes with some forms of natural food (usually frozen or lyophilized). Therefore, flakes **T** and **O** were not mixed, but kept and administered separately, as this is common practice in aquaristics. Two meals per day is a schedule which was suggested in previous research on other ornamental fish [36,41].

Additionally, fish were deprived of food one day per week, which was supposed to resemble a “starvation day”, commonly practiced by hobbyists. The commenced experimental feeding lasted for 12 weeks (time of rearing, *t* = 84 days).

### 2.3. Sampling and Basic Body Parameters

Dead fish were removed daily and final survival rate was calculated. After the trial, all living fish were starved for an additional 24 h, euthanized in MS-222 and weighed (final body weight, FBW). The specific growth rate (SGR, % day^−1^) was calculated from group means of IBW and FBW, using the following formula:SGR = 100 × (ln FBW − ln IBW) × *t*^−1^.

After weighing, fish were sampled for laboratory analyses. For histology, five tetras and five rasboras from each group (including the “Initial” group; 50 fish in total) were fixed in Bouin’s solution for 24 h in 4 °C, followed by 48 h in Surgipath Decalcifier II (Leica Biosystems Nussloch GmbH, Nussloch, Germany) and final flushing with 70% ethanol, in which they were also kept in 4 °C prior to processing. For whole-mount skeletal assessment, the remaining sampled fish (also including the “Initial” group) were fixed in PBS-buffered 4% paraformaldehyde for 96 h in 4 °C, then flushed with distilled water and preserved in 70% ethanol in 4 °C.

### 2.4. Histological Analysis

Bouin-fixed fish were subjected to a standard paraffin embedding procedure, using xylene as the intermediate fluid. Whole fish were sectioned longitudinally using a RM2265 microtome (Leica Biosystems Nussloch GmbH) at 6 µm thickness. Microscope slides were stained with hematoxylin and eosin (HE). Observations and pictures were made using an Eclipse Ni-E microscope, equipped with a DS-Fi3 camera and NIS Elements software (all set parts: Nikon Corporation, Tokyo, Japan).

The nuclear area (NA) and cytoplasmic area (CA) of liver cells was measured for 100 cells per fish (20 hepatocytes × 5 fields of view × 5 fish per feeding regime; group *n* = 500), which is the minimal amount required to neglect the error generated by such planar morphometric approach, as suggested by Rašković et al. [80]. The nucleo-cytoplasmic index (NCI) was calculated separately for each cell using the following formula [71]:NCI = 100 × NA × CA^−1^.

Intestinal fold length (FL) [81,82] was measured for 30 folds per fish (group *n* = 150), which were chosen randomly from the post-pyloric part (neon tetra) or the anterior part (glowlight rasbora) of the guts. The lamina propria was measured from the base to the tip.

### 2.5. Whole-Mount Skeletal Analysis

A double staining protocol for bone and cartilage was applied to the paraformaldehyde-fixed fish, as outlined by Fernández et al. [83,84]. In short, after rehydration, fish were stained overnight for cartilage with non-acidic alcian blue (200 ppm, 80 mM MgCl_2_), similarly followed by alizarin (50 ppm). Afterwards, they were bleached for 3 h (1% KOH, 1.5% H_2_O_2_) and then macerated for 3 weeks (0.25% trypsin, 5% Na_2_B_4_O_7_), and rinsed in dH_2_O. Then, they were stained for bone with alizarin (50 ppm) for 3 days, rinsed in 1% KOH and finally preserved and photographed while immersed in pure glycerin, in 6-well plates. Pictures were taken using a SZ-430T stereomicroscope, with a DLTA6300CMOSSEU3 camera and DLT-Cam Viewer software (Delta Optical, Mińsk Mazowiecki, Poland).

### 2.6. Statistical Analysis

Firstly, the obtained numerical datasets were analyzed for normality using Shapiro–Wilk’s test. For each species separately, the differences in FBW between groups and between the IBW and FBW of each group were analyzed for significance (*p* < 0.05) using the non-parametric Kruskal–Wallis test. Meanwhile, differences in hepatocyte NA, CA and NCI, and intestinal FL were analyzed for significance (*p* < 0.05) using a one-way ANOVA with Fisher’s post-hoc test. All calculations were performed using Statistica v13 (TIBCO Software, Palo Alto, CA, USA). Parameters were displayed as group means ± standard deviation (SD).

## 3. Results

### 3.1. Body Weight, Survival and Other Observations

Basic fish parameters are shown in Table 3. No differences in FBW were found between any of feeding groups of either species. For neon tetras, only group T had a significantly higher FBW when compared to each group’s IBW. Meanwhile, all four groups of glowlight rasboras had a significantly higher FBW than IBW. The SGR was higher in T and TO groups of tetras (than in O and TOL), while the highest SGR for rasboras was found in group TOL. In addition, the average SGR of rasboras in the experiment was almost 40% higher than the SGR of tetras. In terms of survival, pooled group values revealed that tetras had a lower survival rate (70%) than rasboras (92%) within the whole trial.

### 3.2. Histological Analysis

At first notice, histological analysis of livers revealed cases of lipoid degeneration, which occurred (mostly) locally within the parenchyma and were observed with varying frequency throughout the experimental groups, in both species (as illustrated in Figure 1).

The results of morphometric measurements in the livers are displayed in Table 4 (raw data in Appendix A). Similarly for both species, the mean NA reached the significantly highest value in the TOL group, while in TO, it was lower than in both TOL and O groups. In contrary, hepatocyte CA changed differently for each species: the CA in all four groups of neon tetra diminished in comparison to the “Initial” group, likewise reaching the lowest values in TO and TOL groups, while a similar pattern was observed for glowlight rasboras but with a pronounced exception of the TOL group, which showed the highest CA, significantly higher than in the “Initial” group and more than twice as high and CA in the TO group. The NCI had the lowest value in the “Initial” groups of both species, but the highest either in group TOL (tetras) or TO (rasboras).

Exemplary histological pictures of livers (representative of each groups’ mean hepatocyte CA) are given in Figure 2. The most pronounced cytoplasmic eosinophilicity was observed in group T of neon tetras and group TO of glowlight rasboras. Cytoplasmic vacuolization of hepatocytes was noticeable in groups O and TO of neon tetras (group TOL to a lesser extent), while in glowlight rasboras, it was especially visible in group TOL.

Gross observations of the digestive tracts did not reveal any significant pathologies in the experimental fish, in neither species. To support the obtained morphometric data, exemplary histological pictures of measured intestinal folds are displayed in Figure 3.

Morphometric measurements of intestinal FL are displayed in Table 5 (raw data in Appendix A). Mean FL changed differently for each species: in neon tetras, the FL was the significantly highest in group TOL and lowest in TO, while in glowlight rasboras, the FL in group TO was significantly higher from the other, indifferent groups.

### 3.3. Whole-Mount Skeletal Analysis

Upon close examination of the axial skeleton of the double-stained fish, no apparent skeletal deformities were found, neither in neon tetras, nor in glowlight rasboras, with the sole exception of a minor instance of vertebral compression in a rasbora specimen from group T (Figure 4). Exemplary pictures of caudal sections of the skeleton of fish from all experimental groups are shown in Figure 5.

None of the stained fish of either species demonstrated any pronounced signs of skeletal demineralization, as all skeletal elements were thoroughly stained with alizarin, even the tips of fin rays. However, the tissues of glowlight rasboras from the “Initial” group appeared to bind alizarin in a much stronger way than in the four experimental groups, resulting in a dark-purple color (even when using excessive illumination). This occurrence was especially visible when comparing the pictures of the cranium (Figure 6).

## 4. Discussion

Anecdotal observations made by hobbyists and fish producers indicate that both species are relatively mild-tempered, seldomly proceeding with actions such as fin-nipping; thus, they can be safely stocked in community tanks with other small ornamentals. In the discussed experiment, however, there were some behavioral interactions, with neon tetras showing minor acts of both intra- and interspecific aggression, which could have contributed to their lower survival rate. Most of the time, both species were swimming mixed up together, although *P. innesi* usually occupied areas closer to the substrate, while *T. hengeli* remained closer to water surface. In the study by Saxby et al. [85], a two-fold increase in the stocking number of individuals (from five to ten) had a profoundly calming effect on neon tetras and white cloud mountain minnows, *Tanichthys albonubes* (Cypriniformes), while also improving their shoaling tendencies. Albeit the initial fish stocks in our study were in between these numbers (eight per species), it nevertheless will be a reasonable idea to follow such scientifically verified outlines in future research on small ornamentals (especially when studying community aquaria), which means keeping the fish in groups of 10+ individuals of each species. It needs to be considered, however, that larger tanks will likely be required to avoid overcrowding, as this may cause stress and anxiety, and often accelerates the onset of diseases [86].

The final BW of neon tetras was significantly higher than the starting BW only in the T group. Among the few papers on *P. innesi* which could provide a point of reference [29,30,31,62,63,64,65,66,67], it was established that these characids prefer high-protein diets (>50%) [30,31] and that animal protein provides better growth rates than plant protein [30]. This might be a slight suggestion that of the two tested commercial feeds, the protein content/origin of TetraMin flakes was preferred by neon tetras, at least due to its higher total dietary percentage (47.8%, as opposed to 43.3% in the Omega flakes). Unfortunately, this statement should be treated with extreme caution, as the mortalities recorded in each group could have had an effect on the final statistical calculation. On the other hand, the content and ingredients of the two flake diets differed almost completely one from another, which further prevents us from drawing any credible conclusions regarding this matter. More advanced nutritional studies, using precisely formulated diets, definitely need to be conducted to address this ambiguity.

Meanwhile, all groups of glowlight rasboras recorded a significant increase in BW (as evidenced by the higher SGR values), which possibly indicates their younger relative age when compared to the tetras. As no studies were ever performed on *T. hengeli* and only irrelevant, basic feeding evaluations were made for some other species of the Rasborinae subfamily [87,88], the drawing of any further conclusions is severely hindered. Conversely, more and more is known about the nutrition of Danioninae [89], the other popular subfamily of danionids, precisely about their one major representative—the zebrafish. However, in order to avoid deviating too much from the currently discussed issues, we have included an expanded commentary at the end of this section.

All in all, it appears that this simplistic approach of comparing only the BW measurements did not reveal any crucial discrepancies between the tested dietary groups, which was the exact outcome we expected to occur. As we have shown in our previous nutritional experimentation on juvenile crucian carp, *Carassius carassius* [82,90], similar growth rates do not always give the whole picture of the condition of aquarium-reared fish, particularly in regard to the structure of digestive organs or the skeleton. Therefore, additional laboratory analyses (focused on body internals) were performed to confirm or deny whether the studied feeding regimes were truly indifferent in terms of impacting both of the ornamental species.

In consequence, either an imbalance or overflow of dietary fatty acids was likely evidenced in the lyophilizate-fed TOL group of glowlight rasboras by gross histological observations of livers and morphometric measurements of hepatocytes, as emphasized by their significantly larger and little-stained CA, especially in comparison with group TO. In contrary, no such difference between these two groups was found for the neon tetras. In ichthyological research, it is known that the size and degree of eosinophilicity of these cells may be a direct indication of lipid and/or glycogen accumulation [91,92], which happens to be a reversible process [93,94,95], although in the long term, excessive steatosis may result in necrotic changes in the organ [96]. However, in fish aquaculture, lipoid liver degeneration usually coincides with lower growth parameters and arises when fish are fed exclusively on commercial inert diets [97], marking the exact opposite to sparse natural food supplementation which appeared to be the causative factor for this phenomenon in the current trial. In fact, studies on cyprinids: vimba bream, *Vimba vimba* [98] and crucian carp [82], as well as the pike-perch, *Sander lucioperca* (Percidae) [99], all showed that natural food had a predominantly positive effect on hepatocyte structure (size and vacuolization), but note that these experiments were conducted on fast-growing larvae or post-larval juveniles, not on near-adult fish such as the questioned tetras and rasboras.

Nevertheless, it appears to be a quite justifiable presumption that the composition of lipids in the used lyophilizate mixture was highly inadequate for the rasboras, but its administration twice per week might have been simply too excessive, as well. In support of this reasoning, it was revealed that a total replacement of fish oil with vegetable oils in the diet of gilthead sea bream, *Sparus aurata* (Sparidae) resulted in increased hepatic accumulation of fat, as well as early signs of developing lipoid liver disease [100], and similar observations were also made simply for high-fat diets given to this species [91]. The latter report was backed up by a subsequent study on the Wuchang bream, *Megalobrama amblycephala* (Cypriniformes), which yielded similar results [101]. Furthermore, an increase of dietary lipid content (from 5% to 12%) promoted body fat accumulation in the characid fish, *Brycon orbignyanus* [102]. Replacing dietary fish meal with black soldier fly, *Hermetia illucens*, at rates higher than 50% also caused hepatic steatosis in zebrafish [51].

Meanwhile, when looking at other existing studies conducted on fish closely related to the two discussed ornamental species, it was shown that prolonged fasting of the trahira, *Hoplias malabaricus* (Characiformes) [103] and pond loach, *Misgurnus anguillicaudatus* (Cypriniformes) [104], as well as the cyprinids: common carp, *Cyprinus carpio* [105] and tench, *Tinca tinca* [106] caused a significant decrease of their hepatocytic areas, implying the depletion of stored lipids and glycogen. In our trial, starvation was definitely out of the question, but it seems that the combined use of two flake diets in group TO resulted in a more effective nutrient utilization. However, the cytoplasmic eosinophilicity of neon tetra hepatocytes was more pronounced in group T (not in group TO as in the rasboras), especially when compared to the vacuolized hepatocytes in all three groups given the Omega flakes. This implies that the Tetra flakes are probably more adequate for the tetras as they allow for a higher glycogen accumulation with lesser lipid vacuoles. Obviously, specific nutritional studies, oriented towards the preferred dietary lipid content of both species, would have to be performed in order to verify these claims. At least, it can be concluded that all fish have been visibly overfed prior to the experiment (during the quarantine period), as shown by their large hepatocytes in the “Initial” groups.

In addition, the nuclear size and shape of hepatocytes is a supportive indicator of the metabolic condition of the liver, with shrunken, amorphous nuclei implying lower levels of nucleic acid transcription, which in turn leads to impaired hepatic activity due to decreased protein synthesis. Such changes may be elicited by malnutrition [71,107,108,109]. Most noticeably, in both species, the dietary supplementation of the lyophilizate in group TOL significantly improved the NA of hepatocytes (compared to TO), despite the fact that the nuclei of the largest adipocyte-resembling cells in the rasboras were flattened and pushed aside towards the cell membrane due to accumulated lipid vacuoles [82].

Apart from measuring hepatocyte sizes, the intestinal FL is another commonly used a histomorphometric parameter, especially in nutritional studies on cultured fish. Longer folds are naturally a positive growth indicator, as they ensure a larger absorptive area of the mucosa [81,82,110]. There definitely appears to be a convergence between the FL and some of the previously discussed results. In the case of neon tetras, group T was characterized by the significantly longest intestinal folds, mirroring the data for FBW and SGR (only group T had a significantly higher FBW than IBW), while the FL in TO showed the lowest value, similarly to hepatocyte NA and CA (which likely confirms the inadequacy of this regime for *P. innesi*). Meanwhile, only group TO of rasboras distinguished itself from the others in terms of FL, which possibly confirms our previous remarks about this feeding strategy being advantageous for these danionids.

In ornamental fish, it is known that excessive stocking density may increase the risk of body deformation, but only at very high numbers, typically found in zebrafish housing systems of research facilities rather than in regular aquaria or even wholesaler tanks (above 12 fish L^−1^) [86]. Meanwhile, aquaculture research shows that skeletal deformities are relatively frequent in some farmed fish species [73] and can be prompted by a number of reasons [74], although improper nutrition remains to be the main causative factor. In such cases, decreasing mineralization of skeletal tissues may lie at the foundation of these pathologies, since softer bones become more and more prone to twisting and bending forces [111]. This phenomenon begins with insufficient intake/absorption of the three crucial macroelements (Ca, P, Mg) [112], which are then emergently recovered from skeletal tissues for more important physiological purposes, either via osteoclastic resorption [113] or halastatic demineralization [114]. Unfortunately, these processes have not been studied extensively in freshwater fish, but we made such observations in our trial on juvenile crucian carp [90], where the use of commercial feeds significantly decreased bone hardness and resulted in vertebral anomalies. Skeletal double staining was, therefore, conducted in the current study to inspect whether such remarks could be made for housed ornamentals fed exclusively on artificial flakes.

As a result, we did not find any clear signs of such pathologies in the stained fish (only one minor, random incident of vertebral compression). In reality, this was not very surprising, since skeletal deformities mainly occur in rapidly growing, young fish larvae and juveniles [73] (with the exception of salmonids [115]), while the ornamentals used for the purpose of our experimentation were already relatively large, given the size standards of the two species. Thus, it appears that the dietary addition of the lyophilizate was not necessary to sustain the level of skeletal tissue mineralization in grown-out specimens of neon tetra and glowlight rasbora. The latter species, however, probably revealed early signs of demineralization in all four groups when compared to the smaller “Initial” fish group sampled at the beginning of the study. After all, Cypriniformes tend to develop skeletal pathologies when fed exclusively with commercial feeds [90,116,117,118,119,120,121,122,123,124], which possibly could have been happening here, as well. Notwithstanding, such statement requires to be validated in future trials, which should last much longer than three months.

All of this research gently points toward a general conclusion that the two studied species differ from each other in terms of nutritional demands, as presumed in regard to the dietary lipid composition. Even though the addition of natural food usually may have a beneficial effect on liver metabolism, the exact tested lyophilizate did not cause any profound changes when introduced to neon tetras, while being of high risk when given to glowlight rasboras due to the probability of developing hepatic steatosis. Furthermore, it seems that the alternated use of both flake feeds in TO also brought different results for each species, proving beneficial for the rasboras but inappropriate for the tetras, the latter of which had a healthier structure of livers and the longest intestinal folds when fed monotonously with the Tetra flakes. Interestingly, it also appears that the addition of natural food helped to mitigate some of the negative effects of the Omega flakes on *P. innesi*, as shown via histology. Meanwhile, the varied diets did not have any obvious effects on the skeletal structure of the two species, although a slight trend towards demineralization has been observed in all rasboras.

In our study, we highlight the necessity to perform advanced scientific analyses during the exploration of the little understood subject of ornamental fish nutrition. There are simply too many unknowns which need to be addressed, to ensure not only the progress of the industry, but also the improvement of fish welfare (which should be the centerpiece of the puzzle). Diets need to be better matched to the nutritional demands of the fish, as it is clear that even thoughtful, solicitous owners of home aquaria may easily overdo with the feeding of their aquatic pupils, causing potentially irreversible pathologies.

There is some likelihood, however, that awareness among ornamental fish breeders, sellers and hobbyists about different aspects of fish nutrition will be raised, and in a not-so-distant future. The last two decades have seen a near-exponential growth of biological research conducted worldwide on an emerging laboratory species, which also happens to be an ornamental danionid—the zebrafish [125]. Not surprisingly, along with the overabundance of studies in which *D. rerio* is used only as a model organism, more and more published papers focus on the species itself, analyzing and discussing various aspects of breeding and rearing, including nutrition during all life stages, from larvae [18,19,20,21,22,23,24], through juveniles [45,49,51,126,127], up to adults [46,47,48,50,78,128,129,130,131,132]. Obviously, to compare zebrafish husbandry in research facilities to the reality of household or shop aquaria is a far-reaching simplification, at best. It seems plausible, however, that in order to verify the various demands of particular species, this science-based approach could be extrapolated into the ornamental fish industry and hobby, but only as a result of a joint cooperation of practitioners and researchers. If this truly happens, then the emphasis of such studies should likely be placed on the determination of nutritional preferences of small, closely related taxa, simply because conducting such research independently on every single popular aquarium species would prove economically unjustified, as well as ethically unnecessary. In consequence, this approach would lead to the composition of highly dedicated commercial feeds and optimized feeding protocols for different families/genera, definitely improving the welfare of ornamentals kept either privately, by distributors or suppliers.

## 5. Conclusions

Although more expensive and less practical in use than dry feeds, natural food is universally recognized as superior within the global aquaristics community. The results of the discussed experiment may, therefore, surprise many practitioners, as it was shown that while just two such lyophilized planktonic meals a week significantly affected the condition of small omnivorous ornamentals, the results were not exactly favorable as would be originally presumed. Although there were no clear negative outcomes of such supplementation in neon tetras, a highly undesired lipid accumulation was revealed in livers of glowlight rasboras. After all, prolonged hepatic steatosis can have a serious impact on fish health and may even cause untimely mortalities. Measurements of intestinal folds further outlined differences between the two species, with the monodiet consisting of only TetraMin flakes proving superior for the tetras, while the dual-feeding with TetraMin and Omega One yielded the best results for the rasboras. Skeletal analyses further disproved the theory that dietary supplementation with natural food is indispensable for housed ornamentals. In truth, while no attempt was made to assess the exact dietary demands of both species, the study clearly showed that co-housing of ornamental fish can be a challenging task, as applying a balanced feeding regime can be difficult due to hidden biological differences which may distinguish the species within a community tank.

This study also emphasizes profoundly that not only should research-based evaluations of dietary requirements for aquarium fish be conducted on a much more frequent basis (which would allow the formulation of adequate feeds/regimes), but special attention has to be paid obligatorily to the anatomical and physiological features of the species and differences therein. Ultimately, the taxonomic diversity of ornamentals goes far beyond the few common divisions, according to which commercial feed producers prepare most of their products (herbi-, omni-, carnivores; cichlids, guppies, goldfish, bettas, etc.).

## Figures and Tables

**Figure 1 animals-11-03520-f001:**
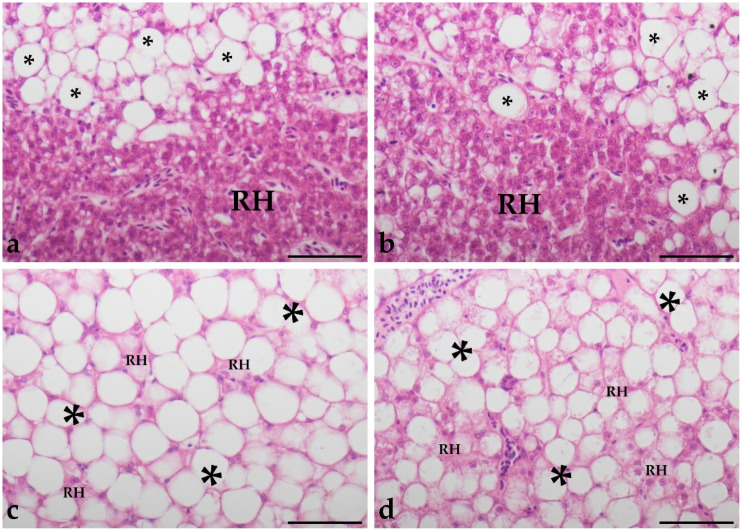
Histological pictures of lipoid degeneration found in liver parenchyma of studied fish. Transition zones between regular hepatocytes (RH) and adipocyte-like hepatocytes (✱) were shown in a specimen of neon tetra from group O (**a**,**b**) and two specimens of glowlight rasbora from group TOL (**c**,**d**). Hematoxylin-eosin stain, scale bars = 50 µm.

**Figure 2 animals-11-03520-f002:**
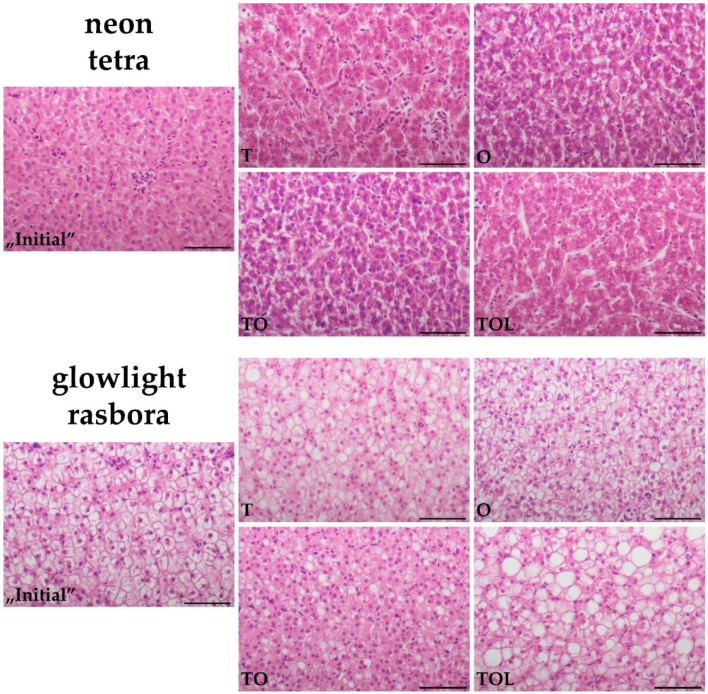
Representative histological pictures of liver parenchyma in each of the studied experimental groups of neon tetras and glowlight rasboras (subpictures named accordingly to group abbreviations). Hematoxylin-eosin stain, scale bars = 50 µm.

**Figure 3 animals-11-03520-f003:**
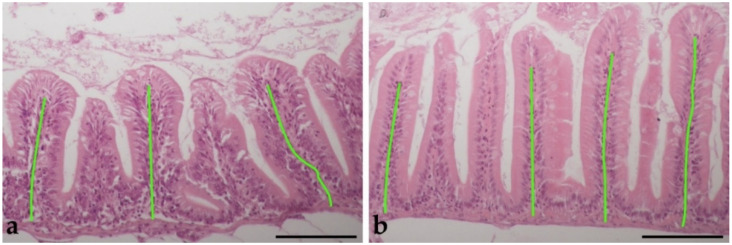
Representative histological pictures from: (**a**) the post-pyloric part of the gut of neon tetras and (**b**) the anterior part of the gut of glowlight rasboras. Green lines depict the intestinal fold length measurements. Hematoxylin-eosin stain, scale bars = 100 µm.

**Figure 4 animals-11-03520-f004:**
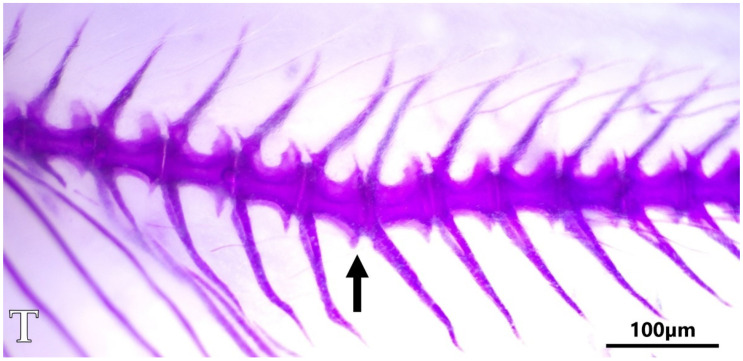
The image of a skeletal deformity found in a specimen of glowlight rasbora from group T. The black arrow points towards an abnormal flexion angle (compression) between the 4th and 5th caudal vertebrae. Whole-mount double staining for bone and cartilage, scale bar = 100 µm.

**Figure 5 animals-11-03520-f005:**
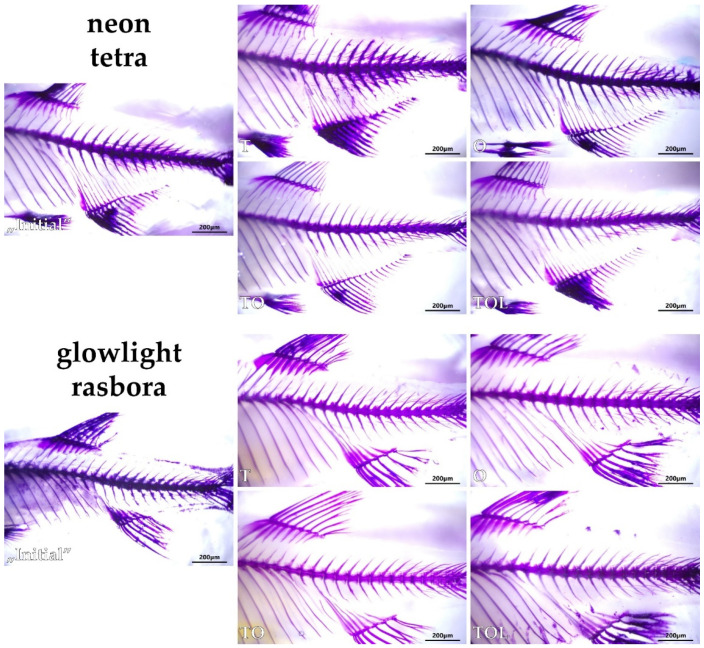
Representative images of caudal sections of the axial skeleton in each of the studied experimental groups of neon tetras and glowlight rasboras (subpictures named accordingly to group abbreviations). Whole-mount double staining for bone and cartilage, scale bars = 200 µm.

**Figure 6 animals-11-03520-f006:**
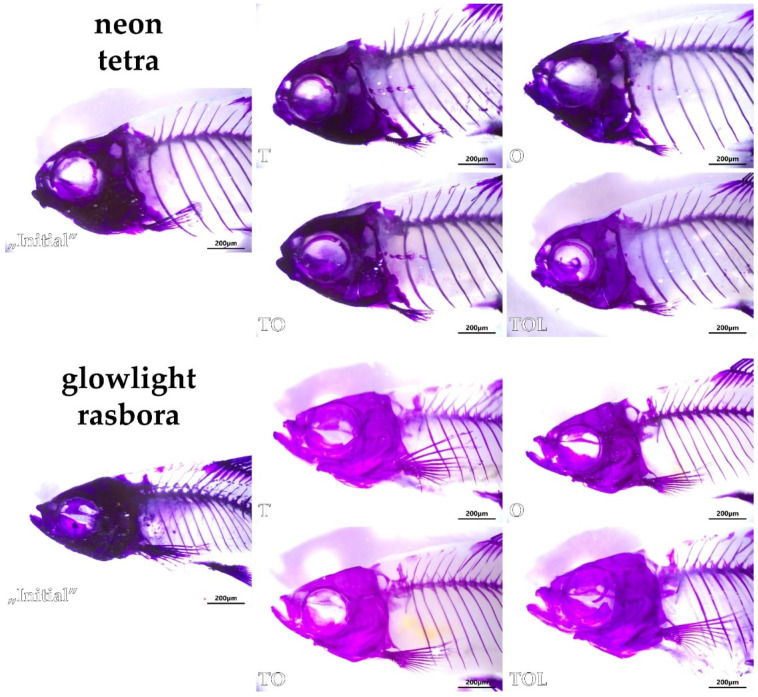
Representative images of the cranium in each of the studied experimental groups of neon tetras and glowlight rasboras (subpictures named accordingly to group abbreviations). Whole-mount double staining for bone and cartilage, scale bars = 200 µm.

**Table 1 animals-11-03520-t001:** Proximate composition and main ingredients of the three feeds used in the study.

	TetraMin Flakes	Omega One Freshwater Flakes	Lyophilizate Mix
Crude protein * (%)	47.8	43.3	36.4
Crude fat * (%)	10.4	11.6	7.7
Crude fiber * (%)	2.7	0.9	5.4
Ash * (%)	6.3	8.0	4.9
Moisture (%)	5.7	6.6	5.2
Main ingredients	Fish and fish derivatives, Cereals, Yeasts, Vegetable protein extracts, Mollusks and crustaceans, Oils and fats, Various sugars (Oligofructose 1%), Algae, Minerals	Salmon (20%), Halibut (16%), Pollock (14%), Herring (13%), Shrimps (11%), Krill (10%), Wheat Flour (9%), Wheat Gluten (4%), Kelp (2%)	Lyophilized: krill (20%), tubifex (20%), brine shrimp (20%), bloodworms (20%), cyclops (20%)
Additives	Vit. A (37,680 IU/kg), Vit. D_3_ (1990 IU/kg)	Vit. A (15,400 IU/kg), Vit. C (0.88 g/kg), Vit. D_3_ (2200 IU/kg), Vit. E (750 IU/kg), Vit. B_2_ (22 IU/kg), Vit. B_3_ (0.11 g/kg), Vit. B_5_ (0.015 g/kg), Vit. B_7_ (0.055 g/kg), Vit. B_9_ (0.08 g/kg), Vit. B_12_ (0.055 IU/kg), Inositol (0.11 g/kg), Astaxanthin (0.6 g/kg), Lecithin (0.5 g/kg), Tocopherol (0.1 g/kg), Ethoxyquin (1 g/kg)	N/A

*****—Calculated as a percentage of dry matter (DM).

**Table 2 animals-11-03520-t002:** Weekly feeding schedule applied for the four experimental groups during the trial period of 12 weeks.

	Monday		Tuesday		Wednesday		Thursday		Friday		Saturday		Sunday	
Meal Time	9:00	16:00	9:00	16:00	9:00	16:00	9:00	16:00	9:00	16:00	9:00	16:00	9:00	16:00
Group T	T	T	T	T	T	T	T	T	T	T	T	T	No meals	
Group O	O	O	O	O	O	O	O	O	O	O	O	O		
Group TO	T	O	T	O	T	O	O	T	O	T	O	T		
Group TOL	T	O	T	O	T	L	O	T	O	T	O	L		

T—TetraMin Flakes; O—Omega One Freshwater Flakes; L—Lyophilizate mix.

**Table 3 animals-11-03520-t003:** FBW, IBW, SGR and final survival of neon tetras and glowlight rasboras in the trial.

		T	O	TO	TOL
Neon tetra	IBW (mg)	275 ± 44	269 ± 48	272 ± 50	274 ± 58
	FBW (mg)	384 ***** ± 55	344 ± 72	371 ± 118	343 ± 47
	SGR (% day^−1^)	0.398	0.289	0.371	0.269
	Survival	12/16 (75%)	9/16 (56%)	11/16 (69%)	13/16 (81%)
Glowlight rasbora	IBW (mg)	225 ± 64	229 ± 61	226 ± 56	228 ± 89
	FBW (mg)	352 ***** ± 86	348 ***** ± 125	359 ***** ± 81	380 ***** ± 159
	SGR (% day^−1^)	0.534	0.497	0.548	0.609
	Survival	16/16 (100%)	15/16 (94%)	16/16 (100%)	12/16 (75%)

Initial body weight (IBW) and final body weight (FBW) were given as group means ± SD. *****—Statistically significant differences (*p* < 0.05) between the IBW and FBW of each group.

**Table 4 animals-11-03520-t004:** Hepatocyte parameters of neon tetras and glowlight rasboras in the experiment.

		“Initial”	T	O	TO	TOL
Neon tetra	NA (µm^2^)	20.76 **^C^** ± 2.81	21.23 **^C^** ± 3.27	22.08 **^B^** ± 4.22	20.91 **^C^** ± 2.96	23.01 **^A^** ± 5.12
	CA (µm^2^)	79.74 **^A^** ± 28.21	55.87 **^C^** ± 23.97	62.32 **^B^** ± 58.42	47.66 **^D^** ± 17.39	45.61 **^D^** ± 20.15
	NCI (%)	28.78 **^D^** ± 9.25	45.32 **^C^** ± 20.50	44.94 **^C^** ± 18.52	49.15 **^B^** ± 17.59	59.01 **^A^** ± 25.02
Glowlight rasbora	NA (µm^2^)	17.23 **^BC^** ± 2.51	17.18 **^BC^** ± 3.22	17.46 **^B^** ± 2.81	16.88 **^C^** ± 2.73	18.13 **^A^** ± 3.59
	CA (µm^2^)	164.98 **^B^** ± 34.77	139.52 **^C^** ± 65.13	113.35 **^D^** ± 31.34	98.95 **^E^** ± 34.41	204.39 **^A^** ± 147.00
	NCI (%)	10.89 **^E^** ± 2.73	14.03 **^C^** ± 4.80	16.41 **^B^** ± 4.68	18.66 **^A^** ± 5.72	12.38 **^D^** ± 6.15

Nuclear area (NA), cytoplasmic area (CA) and nucleo-cytoplasmic index (NCI) were given as group means ± SD. Means with no common superscript letters indicate statistically significant differences between groups (*p* < 0.05).

**Table 5 animals-11-03520-t005:** Intestinal fold length (µm) of neon tetras and glowlight rasboras in the experiment.

	“Initial”	T	O	TO	TOL
Neon tetra FL	157.76 **^C^** ± 56.04	259.03 **^A^** ± 80.83	231.00 **^B^** ± 94.17	171.77 **^C^** ± 71.18	215.55 **^B^** ± 91.66
Glowlight rasbora FL	158.91 **^B^** ± 69.26	157.55 **^B^** ± 77.40	160.77 **^B^** ± 68.53	179.18 **^A^** ± 63.32	159.01 **^B^** ± 53.63

Intestinal fold length (FL) was given as group means ± SD. Means with no common superscript letters indicate statistically significant differences between groups (*p* < 0.05).

## Data Availability

Not applicable.

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
