# Peer review of "Performance of Co-Housed Neon Tetras (Paracheirodon innesi) and Glowlight Rasboras (Trigonostigma hengeli) Fed Commercial Flakes and Lyophilized Natural Food"

_animals, 2021, doi:10.3390/ani11123520_

Round 1

Reviewer 1 Report

please see the attached.

Author Response

Dear Reviewer,

First of all we would like to thank you for you kind and positive evaluation of our work. Below, we have responded to all of the issues you mentioned in your review. However, please be aware that significantly more substantial changes have been introduced to our manuscript, in order to address the major drawbacks highlighted by Reviewer 2.

Line 35: The last sentence in this section needs to be revised as it was not really good conclusion.

Response: The same issue was mentioned by Reviewer 2. We have rewritten this sentence thoroughly. Actually, there are two sentences now.

Introduction: It need to review more research on the nutritional requirement, diet development and feeding strategies on other ornamental fish species.

Response: We have expanded this description in the relevant paragraph by outlining the most important findings which were described in other studies.

Line 50: references are needed.

Response: Following the advice of Reviewer 2, we have shortened the Introduction where possible, so this sentence has been deleted.

Line 62-63: References are needed.

Response: We have reworked this sentence and added proper references in two brackets, to highlight which studies were exactly mentioned.

Method section: It need to re-write to make it clearer in term of experiment design, sample collection, analysis, etc…

Response: We have slightly altered the arrangement and content of several subsections, specifically we created a new subsection 2.3. Sampling and basic body parameters. We hope that this helps with the better understanding of the whole study.

The feeding schedule: why the authors choose that feeding regime for this study? If it based on literature, you need to cite it. If not, please add the text to explain it.

Response: This issue was also mentioned by Reviewer 2. We have added an additional commentary and citations which explain why such feeding regime was chosen for the trial.

Why the authors not calculate the growth rate?

Reponse: We have now added the calculated SGR in Table 3. Due to the relatively small amount of fish in every group, and only a duplicate of tanks per group, we have only added the SGR calculated from the mean IBW and FBW of each group, without SD. We hope that this satisfies the Reviewer's demands.

Reviewer 2 Report

Simple summary

In general, this section is well described.

Abstract

I recommend the authors to change the conclusion “In conclusion, it was confirmed that both species should be fed differently.” This is too general, conclude in terms of your best results.

Introduction

This section is well prepared; however, I would like to recommend the authors to shorter it and focus on the background of digestive physiological studies.

Material and methods

I consider that authors must analyze the real proximate composition of the commercial and live freeze diet.

Change the term “Zero” for “Initial”

Lines 155-156 “group TOL was alternately given the Tetra and Omega flakes, as well as the lyophilizate mix (twice a week, replacing one of the flake meals).”

What was the criterion or reference for replacing one of the flakes in this treatment?

How did you calculate the nutritional balance in the flake mixture and the flake mixture with the freeze-dried mixture, TO, and TOL treatments to keep the amount of nutrients provided to the fish constant?

The experimental design seems to me to be incorrect, since the authors do not really mix the flakes in the TO treatment, and in the TOL treatment they alternate the supplementation of the T and O flakes with the freeze-dried feed (L) once a week, TOL treatment. In this aspect, it is very complicated to explain if the results obtained in the histological analysis were caused by the alternation of the flakes (T or O) or by including a freeze-dried food once a week. Although the authors did control the amount of feed provided to the fish, they did not control the amount of nutrients (proteins, lipids, and carbohydrates) provided to the fish, which is a problem to explain the differences between treatments in the histological analysis.

Results

This section is well described.

Change the term “Zero” for “Initial”.

Discussion

This section is well structured and covers the importance of proper nutrition in ornamental fish species; however, as non-specific commercial diets are used in the feeding of both fish species, it is not a study that can be analyzed specifically on nutritional requirements, so this section is speculative, as each of the commercial feeds is very different in chemical composition (which should be analyzed) and in types of ingredients used for their manufacture. From the point of view of aquaculture nutrition, it is very important to develop specific feeds through studies of digestive morphophysiology (histology, bone development, digestive enzyme activity, genomic expression, in vitro digestibility, among others), in order to generate knowledge about the nutritional requirements of the species for each stage of development and from there select the best ingredients for the elaboration of commercial feeds, which should be studied in terms of in vivo digestibility.

Conclusion

This section needs to be rewritten in terms of their results according to the limitations of your study.

Author Response

Dear Reviewer,

We are grateful for the critical review of our manuscript. We believe that the quality of our paper improved significantly after applying all of the necessary corrections, which you have suggested. Specific responses can be seen below.

Abstract: I recommend the authors to change the conclusion “In conclusion, it was confirmed that both species should be fed differently.” This is too general, conclude in terms of your best results.

Response: As the same issue was mentioned also by Reviewer 1, we have rewritten this sentence, accordingly. Currently, there are two longer and more elaborate sentences.

Introduction: This section is well prepared; however, I would like to recommend the authors to shorter it and focus on the background of digestive physiological studies.

Response: We have attempted to shorten this section down and, as a result, we deleted an expendable paragraph and several sentences, here and there. However, following the remarks of both Reviewers, we expanded on the background of nutritional research by specifying the main results of some studies which had been conducted on ornamental fish. We believe that these additions further explain the aims of our study.

Material and methods: I consider that authors must analyze the real proximate composition of the commercial and live freeze diet.

Response: To satisfy the Reviewer's demands, we have gone to great lengths to get these necessary analyses performed as rapidly as possible, within a week. The updated parameters are shown in Table 1 and we added a short description about the AOAC methods.

Change the term “Zero” for “Initial”

Response: We have changed the coding of this group throughout the entire manuscript, including the Figures.

Lines 155-156 “group TOL was alternately given the Tetra and Omega flakes, as well as the lyophilizate mix (twice a week, replacing one of the flake meals).” What was the criterion or reference for replacing one of the flakes in this treatment?

Response: We have added several sentences which explain this issue. Generally, we attempted to mirror the habits of hobbyists, who usually enhance the diet of their aquarium fish with natural food, but only from time to time. Instead of feeding the fish with flakes/pellets, they will simply swap one of the meals by giving natural food, of whatever type (live, frozen, lyophilized). As shown in Table 2, every week in the TOL group, the Tetra flakes were replaced once (on Saturdays), as were the Omega flakes (on Wednesdays). Therefore, both flakes were administered equally (a total of 60 meals each in this group).

How did you calculate the nutritional balance in the flake mixture and the flake mixture with the freeze-dried mixture, TO, and TOL treatments to keep the amount of nutrients provided to the fish constant?

Response: We have maintained an almost constant feeding rate (% of BW) for each group, what is still a relatively common practice in nutritional studies on fish, especially in early studies such as ours. Many of the studies conducted on zebrafish, which we cited in our paper, followed the exact same concept. Due to the minimal amount of feed which was administered into each tank at every feeding, and accounting for the growth of the fish, we have indicated the feeding rate as "3-4%". This is still a step ahead of some studies which follow the concept of simply feeding ad libitum.

The experimental design seems to me to be incorrect, since the authors do not really mix the flakes in the TO treatment, and in the TOL treatment they alternate the supplementation of the T and O flakes with the freeze-dried feed (L) once a week, TOL treatment.

Response: We explained the non-mixing issue in the manuscript itself (mixing flakes is a rather uncommon practice, as hobbyists will rather keep their feeds separately, in their original packaging). Furthermore, we did not "supplement" the T and O flakes with the lyophilizate, but we replaced them entirely on those two meals per week. The % BW feeding rate remained constant. We have altered this description slightly to avoid any misconceptions and we also added letter coloring in Table 2, for better visibility of the meals.

In this aspect, it is very complicated to explain if the results obtained in the histological analysis were caused by the alternation of the flakes (T or O) or by including a freeze-dried food once a week. Although the authors did control the amount of feed provided to the fish, they did not control the amount of nutrients (proteins, lipids, and carbohydrates) provided to the fish, which is a problem to explain the differences between treatments in the histological analysis.

Response: Obviously, we did not calculate the amount of nutrients as it was not our aim to address the specific nutritional demands of each species. We wanted to raise awareness that co-housing and co-feeding of ornamental fish can be a relatively complicated issue. Without the consciousness about the significant biological differences between kept species, practitioners might severely affect the welfare of their pets without even knowing it. This was the main purpose of our study concept, as there were no studies before which would address the problem of co-housing of biologically diverse species. All in all, this was not a typical aquaculture study, with perfectly formulated and evened-out, balanced diets. This is why we did not even attempt to submit this paper to any of the known aquaculture-oriented journals.
In regard to the commentary about histological analyses, we would like to emphasize that our major concept was to evaluate the differences between the monotonous T and O feeding groups and the mixed TO group, to see whether there was an improvement or not (if the feeds complemented each other, or one of them was simply more adequate, and for which species). Then, the TOL group was supposed to show whether there could be an even further improvement. However, the outcome was quite different, with the lyophilizate not showing many effects on the tetras, but proving quite harmful for the rasboras (at least in the regime which was applied in our study).

Results: Change the term “Zero” for “Initial”.

Response: As mentioned above, we corrected this issue throught the manuscript.

Discussion: This section is well structured and covers the importance of proper nutrition in ornamental fish species; however, as non-specific commercial diets are used in the feeding of both fish species, it is not a study that can be analyzed specifically on nutritional requirements, so this section is speculative, as each of the commercial feeds is very different in chemical composition (which should be analyzed) and in types of ingredients used for their manufacture. From the point of view of aquaculture nutrition, it is very important to develop specific feeds through studies of digestive morphophysiology (histology, bone development, digestive enzyme activity, genomic expression, in vitro digestibility, among others), in order to generate knowledge about the nutritional requirements of the species for each stage of development and from there select the best ingredients for the elaboration of commercial feeds, which should be studied in terms of in vivo digestibility.

Response: Of course, we fully agree with this reasoning. Aquaculture is our main research area and we are aware of the concepts and basics of such nutritional studies. However, as we already outlined above, this is not an aquaculture nutritional study. We never aimed to establish specific nutritional demands for the studied species, as this might be the target of future experiments, with formulated diets. In terms of science, the world of aquaristics is generally ignored worldwide, therefore, there always needs to be a start. However, following the Reviewer's remarks, we have attempted to tone down some of our conclusions, as we agree that they were mostly speculative.

Conclusion: This section needs to be rewritten in terms of their results according to the limitations of your study.

Response: We have added some improvements according to this and the previous remark about the Discussion.

Round 2

Reviewer 2 Report

I have read your article, and I note that you have made a great effort to improve the focus of the article, which I think is appropriate, mainly because you eliminated the references that have to do with nutritional requirements since your study is not directly related to this type of research.

I recommend that you review the numbering of your references.

Author Response

Dear Reviewer,

We have formatted our references using Mendeley and using the "Animals" template, but indeed, there were some minor mistakes which were fixed with the use of the "Refresh" button in the MS Word Mendeley plugin.

Everything should be correct now.